# Intention Reasoning for User Action Sequences via Fusion of Object Task and Object Action Affordances Based on Dempster–Shafer Theory

**DOI:** 10.3390/s25071992

**Published:** 2025-03-22

**Authors:** Yaxin Liu, Can Wang, Yan Liu, Wenlong Tong, Ming Zhong

**Affiliations:** State Key Laboratory of Robotics and System, Harbin Institute of Technology, Harbin 150001, China; liuyaxin@hit.edu.cn (Y.L.); 22s130227@stu.hit.edu.cn (C.W.); 21b908088@stu.hit.edu.cn (Y.L.); 22s130223@stu.hit.edu.cn (W.T.)

**Keywords:** intention reasoning, object affordance, CP-logic encoding, D-S theory

## Abstract

To reduce the burden on individuals with disabilities when operating a Wheelchair Mounted Robotic Arm (WMRA), many researchers have focused on inferring users’ subsequent task intentions based on their “gazing” or “selecting” of scene objects. In this paper, we propose an innovative intention reasoning method for users’ action sequences by fusing object task and object action affordances based on Dempster–Shafer Theory (D-S theory). This method combines the advantages of probabilistic reasoning and visual affordance detection to establish an affordance model for objects and potential tasks or actions based on users’ habits and object attributes. This facilitates encoding object task (OT) affordance and object action (OA) affordance using D-S theory to perform action sequence reasoning. Specifically, the method includes three main aspects: (1) inferring task intentions from the object of user focus based on object task affordances encoded with Causal Probabilistic Logic (CP-Logic); (2) inferring action intentions based on object action affordances; and (3) integrating OT and OA affordances through D-S theory. Experimental results demonstrate that the proposed method reduces the number of interactions by an average of 14.085% compared to independent task intention inference and by an average of 52.713% compared to independent action intention inference. This demonstrates that the proposed method can capture the user’s real intention more accurately and significantly reduce unnecessary human–computer interaction.

## 1. Introduction

The WMRA is a commonly used form of an assistive robot [1]. However, the traditional joystick remote control mode of the WMRA requires frequent limb movements from the user, which can lead to both physical and psychological burdens. As a result, current research focuses on enabling users to interact with the WMRA with fewer limb movements [2] and convey their intentions with minimal effort.

In most existing research, intent recognition typically involves inferring intentions by analyzing behavior. These studies primarily rely on observing human posture as contextual information to deduce intentions, representing a direct approach to human–environment interaction. However, this method is generally designed for individuals with intact physical abilities. For instance, Ashesh Jain et al. successfully recognized human intentions by analyzing the spatiotemporal structure of motion using Recurrent Neural Networks (RNNs) [3]. Liu et al. combined ST-GCN-LSTM (Spatial Temporal–Graph Convolutional Networks–Long Short-Term Memory) and YOLO models to infer intentions based on changes in human joint movements and object-handling sequences [4]. Similarly, Ding et al. proposed a real-time motion intent recognition method based on Long Short-Term Memory (LSTM) networks for dynamic wearable hip exoskeletons in this paper [5]. Song et al. employed a CNN-RF (Convolutional Neural Network–Random Forest) hybrid model to recognize five types of actions, including standing, sitting, walking, and climbing stairs [6]. Furthermore, Wang et al. presented an offline training and action intention recognition method based on Long Short-Term Memory networks, capable of identifying four sub-action intentions: reach, move, set down, and manipulate [7]. Wang et al. utilized a Three-Dimensional Convolutional Neural Network (3D CNN) to recognize human action intentions frame by frame in video streams [8]. Zhang et al. innovatively transferred visual language models (VLMs) from the image domain to the video domain for Human Action Recognition [9]. Additionally, some studies have integrated human actions with object category cues to predict users’ intentions [10,11].

However, individuals with physical disabilities cannot directly interact with their environment and instead rely on robots for assistance. In such scenarios, assistive robots must accurately recognize users’ intentions and autonomously perform activities of daily living (ADL). In recent years, researchers have explored the concept of “shared attention” [12] to simplify human–robot interaction and reduce the complexity of robot manipulation. Shared attention can be established through various methods, including screen tapping [13], eye gaze [14], laser pointers [15,16], and electroencephalogram (EEG) recognition [17]. Based on these methods, assistive robots infer task intentions by focusing on the “selected object”. For example, Li et al. inferred the user’s intention by analyzing the objects they gazed at and their position using a Naive Bayes graphical probability model [18]. Gao et al. proposed the Neural-Logical Reasoning Network (NLRN) to enhance explicit reasoning capabilities, demonstrating the potential of neural-logical integration in intent recognition [19]. Smith et al. demonstrated how to combine logical reasoning and probabilistic models using ProbLog for intent recognition, providing a new perspective for solving complex intent recognition problems [20]. Zhongli Wang et al. proposed a novel logic framework based on affordance segmentation and logic reasoning for robot cognitive manipulation planning [21]. Xu et al. presented a new framework, LKLR, that combines large language models (LLMs) and knowledge graphs (KGs) for collaborative reasoning [22]. Thermos et al. proposed a dual encoder–decoder model for joint affordance reasoning and segmentation, offering a new approach to understanding human–object interactions [23]. Kester Duncan et al. developed an object action probabilistic graph model network, identifying and learning human intentions by observing objects in the scene, associated actions, and human interaction history. However, this framework is limited to recognizing implicit intentions for individual objects [24]. Liu et al. further considered the relationship between objects and actions by using objects as contextual information to infer the implicit action intentions between multiple objects [25]. Although these studies focus on inferring users’ intentions through scene recognition and objects in the environment, they fail to ensure that the inferred actions are physically feasible for specific objects from the perspective of the object’s functionality.

Nowadays, the concept of affordance—originally proposed by American psychologist James J. Gibson in 1977 [26]—is now widely used in the field of robotic vision to analyze the action affordances of objects. The theory describes the possibilities for use or interaction that objects or environments offer to individuals. Although the shapes and appearances of objects in the real world vary widely, humans can still easily recognize their functions in a short amount of time, even if they have never seen these objects before. For example, the sharp edge of a blade provides the function of cutting, while its handle provides the function of grasping. Martijn et al. drew inspiration from the concept of object affordance and achieved the recognition of the current assembly action and the prediction of the next assembly action based on the sequence of objects operated by the user in a video [27]. Isume et al. utilized affordance prediction to select the best available parts for a craft assembly task, enabling the completion of a full craft assembly task [28]. Hassanin et al. reviewed affordance theories, highlighting that studying object affordances helps predict future actions, identify activities of agents, recognize object functions, understand social contexts, and reveal hidden object values. [29]. Mandikal et al. embedded the concept of object affordance into a deep reinforcement learning loop to learn grasping policies preferred by humans [30]. Deng et al. created a dataset consisting of 18 executable actions and 23 types of objects, which aids robots in identifying the implicit grasping actions of objects [31]. Xu et al. studied methods for expressing affordance, analyzed the long-term execution effects of objects in tasks, and predicted the actions to be performed in the next step [32]. Borja-Diaz et al. proposed a novel approach that extracts a self-supervised visual affordance model from human-teleoperated play data and leverages it to enable efficient policy learning and motion planning [33].

Additionally, some studies have integrated human actions with object category cues to predict users’ intentions. Long et al. further extended the application of affordance in robotic grasping, proposing a novel caging-style gripper system that combines one-shot affordance localization and zero-shot object identification. This system relies solely on scene color and depth information, similar affordance images, and brief textual prompts to achieve flexible grasping without extensive prior knowledge [34]. Do, Thanh-Toan et al. extended grasp affordance from simple robotic grasping to more complex human–object interactions, supporting reasoning for various affordance tasks such as contain, cut, and display [35]. Sun et al. constructed an object-to-object affordance model, making the learned affordances beneficial for robot operations involving multiple objects [36]. Girgin et al. proposed a Multi-Object Graph Affordance Network that models complex compound object affordances using graph neural networks, leveraging depth images and graph convolution operations to predict the outcomes of object–compound interactions and enabling task planning for multi-object interaction sequences (e.g., stacking, inserting, and passing through) [37]. Mo et al. studied the affordance relationships between objects and used implicit attributes of objects to predict the execution modes of four household tasks [38]. Uhde et al. combined human demonstrations and self-supervised interventions to learn the causal relationships between object properties and object affordances, enabling the transfer of learned affordance knowledge to unseen scenarios for effective action reasoning [39].

Current research on action reasoning based on object affordance detection is essentially a classification problem. However, existing studies typically treat it as a single-label classification problem [21,27,35,40,41]. In reality, the affordance of objects is diverse. For example, a chair can not only be used for sitting but also as a footstool or a temporary surface for placing objects. Our method adopts this concept, utilizing deep learning techniques to learn rich features from data. Unlike single-label classification, we indirectly achieve multi-label classification by dividing objects into different functional parts and associating them with multiple primitive actions.

In this study, we propose an innovative intention reasoning method for users’ action sequences by fusing object task and object action affordances based on D-S theory. Specifically, the contributions of this method are summarized below:The task reasoning module in the algorithm employs CP-Logic to model and infer the relationship between object categories and task intentions. Additionally, a task probability update algorithm based on reinforcement learning is developed, enabling the model to adapt to users’ operational habits and achieve object-to-task intention reasoning.The action reasoning module does not rigidly define the functional parts of an object or the actions associated with them as fixed or singular. It also does not predict subsequent action intentions solely by visually detecting an object’s functional regions. Instead, we utilize CP-Logic to probabilistically model the relationships between the functional parts of an object and their potential actions, capturing the inherent flexibility and variability in object action affordances.We incorporate D-S theory to fuse information from the aforementioned reasoning modules, enabling the inference of action sequence intentions for target objects. This approach imposes task constraints on action reasoning, enabling more accurate and reliable prediction of operational intentions, allowing the WMRA to accurately understand users’ task intentions and, during the execution of real-world tasks, select and execute appropriate actions on functional regions of objects in a task-oriented manner.

The rest of this paper is as follows. Section 2 describes our intent reasoning framework. Section 3 describes the specific intention reasoning method. Section 4 reports the experiments and results. Section 5 reports the conclusion.

## 2. Intent Reasoning Framework

As shown in Figure 1, this paper proposes a general framework for the WMRA intent inference model. The model first identifies the objects of the user’s attention using laser interaction, then infers the user’s subsequent task intent and action intent. Among them, how to accurately and reliably reason about the subsequent task intent based on the objects of the user’s attention and reliably execute the task is the key issue in this research. In our previous research, we attempted to utilize the conditional random field (CRF) for inference of implicit object task intent [25]. However, this method only considered the inference of a single task intent for user-focused objects and did not take into account the functional attributes of the objects, as well as the subsequent execution of the task intent, in the intent inference process.

For this reason, this paper improves the algorithm for task intent inference in the WMRA robot implicit interaction system. As shown in Figure 1, after the user pays attention to and selects an object in the scene by laser pointing the object or other interaction means, the system first recognizes the category of the object through the Object Recognition Module and inputs it into the Object Task Affordance Reasoning Module based on CP-Logic encoding. This module combines the user’s historical living habits and the object categories to reason about the task intent.

At the same time, given that different regions of an object have different geometrical forms and action functional attributes, the system performs instance segmentation of the functional regions of the object through the Object Affordance Region Instance Segmentation Module and inputs the recognized functional regions into the Object Action Affordance Reasoning Module. This module is based on CP-Logic encoding and associates the functional regions of the objects with possible subsequent actions.

In order to reason the tasks and actions of objects more accurately, this paper further proposes an inference method for action sequence intent using D-S theory fusing task intent and action intent constraints. The generated action sequence guides the robot in object manipulation under the constraints of tasks, actions, and objects. Our method effectively integrates user operating habits with the action-specific affordances of an object’s functional parts, thus significantly improving the accuracy of task intent inference and the reliability of action execution. Next, this paper will introduce the method in detail.

## 3. Methods

### 3.1. Object Task Affordance Reasoning Based on CP-Logic Encoding Principles

#### 3.1.1. Object Recognition

When a user focuses on a particular object, the WMRA robot system first needs to visually perceive the object the user is focused on and simultaneously identify the object’s class. We use a laser pointer to convey the user’s attention to an object; when the user points a laser pen at an object, it signifies that the user is focusing on that object. Therefore, accurate detection of the laser spot is crucial for the WMRA to focus on the target object. We use YOLOv8 to detect the laser spot, identify the user’s focused object, and recognize the object’s name.

To improve the precision of laser spot detection using YOLOv8, we added an additional 10,000 laser spot images from different home environments to the object recognition dataset and applied data augmentation techniques, including random brightness enhancement, random rotation, and the addition of salt-and-pepper noise. Considering that due to limited physical movement abilities, the laser spot may wobble and cause misreading, we also included samples with missed operations in the dataset to enhance detection accuracy and robustness. More details on laser interaction can be found in previous research [25,42].

#### 3.1.2. Object and Task Ontology Construction

Ontology is a formal description of the concepts and their relationships within a domain. In this study, the purpose of constructing the ontology is to enable the WMRA system to understand the nature and relationships of different objects and tasks in the household and disability-related living environments, thus providing more personalized services and assistance to users.

The International Classification of Functioning, Disability, and Health (ICF) [43] guidelines describe the essential tasks required for maintaining the independence of people with disabilities in a home environment. Therefore, we have constructed a knowledge base by extracting the objects and tasks involved from the ICF.

Object Ontology Knowledge Construction: we focus on four types of objects:(1)Containers that can hold objects in the home environment;(2)Tools that can be used by the WMRA to complete specific tasks;(3)Furniture commonly used in the home environment to place objects;(4)Controllers used to operate the switches of various devices in life.

For objects of the container type, we further classify them into open containers and closed containers, which can be described through the object ontology shown in Figure 2a. We constructed five object container classes, two furniture classes, two controller classes, and four tool classes. Of course, based on the requirements, more object bodies can also be constructed. The type of each object is determined by its attributes and common knowledge. Using CP-Logic [44], we convert object ontology knowledge into deterministic logical rules, where deterministic logic specifies whether an object belongs to a container class or another class. For example, *furniture(O) ← chair(O)* means if the object O is a chair, it belongs to the furniture class.

Task Ontology Knowledge Construction: We focus on nine common tasks in daily home life, denoted as *T = {pass, use, pourIn, pourOut, grasp, place, push, press, insertIn}*, as shown in Figure 2b. For example, the rule *pour(T) ← pourOut(T)* means that a task involving pouring an object out of a container is classified as a pour task.

With this ontology knowledge, we can reason through the object ontology to determine whether the object of interest belongs to the container class or another class. Additionally, using task ontology knowledge, we can identify the type of task that can be performed on the object. Furthermore, to determine the user’s task intent for the object of interest, it is necessary to establish a constraint relationship between the object and the task, i.e., to establish an object task affordance and encode it probabilistically.

#### 3.1.3. Object Task Affordance Construction and CP-Logic Encoding

There exists a certain constraint relationship between tasks and objects, and the robot can infer the user’s task intention based on the object the user is focused on. For example, a person can pour water into a cup, which links the cup to the pouring task.

However, a knife cannot be associated with the pouring task, but it can be associated with the cutting function and thus linked to the using task. After extracting the object ontology and task ontology information from the real-world scenario, we establish object task (OT) affordance to represent the constraints between objects and tasks, as shown in Table 1.

OT affordance allows us to link objects and tasks together and helps us define a task intention reasoning model in a relational manner. The constraint relationships between tasks and objects are defined by human experience. For instance, a knife cannot be used for a pouring task, and a container with a small opening is difficult to pour liquid into. However, some people might think that a small-opening container is easier to pour liquid from. In general, the constraints between objects and tasks consider the habits of most people while ignoring individual preferences and lifestyle differences.

CP-Logic is a logical framework that combines causal relationships and probabilistic reasoning. It represents and infers uncertainty and causality by introducing causal rules and probabilities into logical programs. The logical rule is shown as follows:(1)p::β←α

It represents the probability *p* of event *β* occurring under condition *α*.

Due to external environmental and human factors, the robot’s understanding of the relationship between objects and tasks should be probabilistic rather than deterministic. To more reasonably encode the constraints between objects and tasks, we use CP-Logic to encode OT affordance, which also balances the cognitive differences among different users. The size of the probability value measures the degree of relevance between objects and tasks in the eyes of different users. For example, the rule *0.9::Task(X, use, O) ← robot(X), tool(O)* indicates that if the object the user is focused on belongs to the tool class (tool), the inferred user task intention is to perform the use task, with a probability of 0.9. The probability value influences the relevance between the object and the task: the higher the probability, the stronger the relevance; the lower the probability, the weaker the relevance.

We not only consider the user’s focus on a single object but also infer task intentions involving multiple objects. We believe that the order in which the user focuses on objects implicitly indicates the sequence of object operations. For example, the rule *0.93::Task(X, pourOut, O_1_, O_2_) ← robot(X), canister(O_1_), openContainer(O_2_)* indicates that if the user first focuses on object *O_1_* (which belongs to the canister class) and then on object *O_2_* (which belongs to the open container class), the inferred user task intention is to perform the *pourOut* task from object *O_1_* to object *O_2_*, with a probability of 0.93. Some examples of mapping OT affordance to probabilistic logic rules are shown below:


*0.7::Task(X, press, O) **←** robot(X), controller(O)*


*0.4::Task(X, insertIn, O**_1_, O**_2_)*
 *←* 
*robot(X), tool(O*
*_1_), openContainer(O*
*_2_)*

#### 3.1.4. User Habit Adaptation Based on Reinforcement Learning

In the previous sections, we first used the object recognition network to obtain the name of the object the user is focused on, then established the object/task ontology and object task affordance, and encoded the object task affordance into probabilistic relations based on CP-Logic to build the object task intention reasoning model.

To allow the reasoning model to gradually learn the user’s habits over time as they use it in daily life, we introduced reinforcement learning. Based on the user’s feedback on the inferred task intentions (whether they accept or reject the task intention), we dynamically adjust the priority of the reasoning tasks. Tasks that are accepted multiple times by the user are recommended with higher priority, while tasks that are rejected multiple times are recommended later.

We interact with the user by presenting the inferred task intentions and continue interacting until the user accepts a specific task intention. We define this entire process as a time-step in the user’s daily life choices and record all feedback from interactions during the session. Figure 3 illustrates the overall process of task intention reasoning, user interaction, and habit adaptation.

We employ a weighted average strategy based on historical feedback to dynamically adjust the encoded probabilities of object task affordances. This strategy combines current and historical feedback, as shown in the following Equation (2):(2)$$p(OT)\leftarrow (1-\alpha )\cdot p(OT)+\alpha \cdot \frac{\sum_{i=1}^{n}w_{i}\cdot F_{i}}{\sum_{i=1}^{n}\;w_{i}\;}$$
where

*α* is the learning rate, controlling the weight between current and historical feedback.

*w_i_* is the weight of the *i*-th feedback, typically set using a time decay.

*F_i_* is the *i*-th feedback.

### 3.2. Object Action Affordance Reasoning Based on Visual Affordance Detection

Inferring implicit task intentions based on object names and categories is a reliable method. However, for the WMRA, during actual task execution, the robot does not know what specific actions to take or which functional part of the object the action should be applied to. This makes it difficult for the robot to effectively complete the task based on inferred task intentions. Inspired by affordance labels [45], we directly associate seven affordance labels—*grasp, wrap–grasp, cut, scoop, contain, pound, and support*—with different functional parts of the object. In reality, functional parts of objects can serve multiple purposes. For example, the body of a cup can afford not only actions like *wrap–grasp* and *scoop* but also the function of containing objects. Therefore, we first segment the object into different functional parts. Then, based on actual usage scenarios, we associate these functional parts with one or more atomic actions to achieve more accurate action reasoning.

#### 3.2.1. Segmentation of Object Functional Regions

In this study, we focus on eight functional parts of an object, defined as *Parts = {holdingPart(O), poundingPart(O), cuttingPart(O), scoopingPart(O), containPart(O), buttonPart(O), brushPart(O), supportPart(O)}*. To segment an object into its functional parts, we use the classic instance segmentation model, Mask R-CNN. Mask R-CNN is a powerful instance segmentation algorithm that simultaneously performs object detection and pixel-level segmentation. Its end-to-end training enables learning object classification, bounding box regression, and segmentation without extra post-processing. Additionally, Mask R-CNN is adaptable, allowing for easy integration with different backbone networks (e.g., ResNet) for enhanced performance. It strikes a balance between accuracy and efficiency, making it suitable for a variety of applications such as autonomous driving, medical imaging, and video surveillance.

#### 3.2.2. Object Action Affordance Construction

Affordance theory describes the possibilities for use or interaction that an object or environment offers to an individual. For instance, the sharp edge of a blade affords cutting, while its handle affords grasping. However, there is no one-to-one correspondence between an object’s functional parts and atomic actions. For example, the sharp edge of a blade can be used for cutting or grasping to hand the blade to someone else.

Once the functional parts of an object are identified, we leverage the concept of affordances to associate these functional parts with specific actions, thereby defining object action affordance (OA). This serves to constrain the relationship between the object’s functional parts and the actions performed, forming the basis of a functional part-action intention inference model. The constraints are summarized in Table 2. In this study, we focus on ten atomic actions: *Action = {grasp, pourWith, placeOn, push, press, cutWith, poundWith, brushWith, scoopWith, insertInTo}*.

Similarly, the robot’s understanding of the relationship between an object’s functional parts and atomic actions should be probabilistic. Initial probabilities are assigned based on the geometric features of the functional parts and practical functional constraints. For instance, the sharp edge of a blade has a high association with the “*cut*” action and a lower association with the “*grasp*” action. Example rules are as follows:


*0.95:: cutWith(X, O, cuttingPart(O)) ← Object(O)˄cuttingPart(O)*



*0.3:: grasp(X, O, cuttingPart(O)) ← Object(O)˄cuttingPart(O)*



*0.1:: push(X, O, cuttingPart(O)) ← Object(O)˄cuttingPart(O)*


However, this approach has some limitations. For instance, focusing solely on an object’s individual functional parts without considering the connections between different parts may lead to action intentions that do not align with user expectations. Additionally, when inferring action intentions for multiple objects sequentially attended to by the user, the reasoning may simply generate combinations of actions derived from functional parts without task-specific constraints, making it challenging to identify the true action intention. In the following, we will introduce intention reasoning for user action sequences through the fusion of object task and object action affordances based on D-S theory.

### 3.3. Action Sequence Intention Inference Based on D-S Theory

To more accurately infer user intent, we explore the fusion of object task and object action affordance. In this work, we utilize D-S theory to reason about both task intentions and action intentions under the dual constraints of object task and object action affordance. This approach generates a sequence of action intentions for the objects, enabling the WMRA to not only accurately understand the user’s task intentions but also execute appropriate actions on the functional parts of objects in a task-oriented manner during actual operations.

#### 3.3.1. Introduction to D-S Theory

Dempster–Shafer Theory [46,47,48], also known as Evidence Theory or Theory of Belief Functions, is a mathematical framework for handling uncertain information. In D-S theory, a set of fundamental events is referred to as the discernment frames, denoted as *Θ*. The events within the discernment frames are mutually exclusive execution events, expressed as *Θ* = {*θ*₁*, θ*₂*, …, θₙ*}. The power set of the discernment frames is represented as 2*^Θ^* = {*A*: *A* ⊆ *Θ*}, which includes all possible subsets. In the discernment frames, The function *m*(*A*), which represents a Basic Probability Assignment (BPA), is a mapping 2*^Θ^* → [0, 1] such that *m*(∅) = 0 and $\sum_{A\subseteq \Theta }\;m(A)=1$.

The BPA measures the degree of support assigned to proposition *A* ⊆ Θ. Subsets with non-zero probability mass are referred to as focal elements and form a set F. A Body of Evidence (BoE) is represented as the triplet {Θ,F,m(⋅)}. Given a BoE, the belief function for a set *A* is defined as $Bel(A)=\sum_{B\subseteq A}\;m(B)$. The belief function quantifies the degree of trust in proposition *A*. The plausibility function for a set A is defined as $PI\left(A\right)=1-Bel\left(\bar{A}\right)$, where $\bar{A}$ is the complement of *A* in the discernment frames *Θ*. The plausibility function *PI(A)* incorporates the basic belief of all sets compatible with *A*. The belief interval for *A*, expressed as [*Bel(A)*, *PI(A)*], indicates the degree of confirmation for a given hypothesis. Additionally, the confidence measure *μ*, as defined in [49], is employed to compare the uncertainties associated with the proposition δ and its corresponding uncertainty interval [*Bel(δ)*, *PI(δ)*], shown in Formula (3):(3)$$\mu (Bel(\delta ),PI(\delta ))=1+\frac{PI(\delta )}{\nu }\;log_{2}\frac{PI(\delta )}{\nu }\;+\frac{1-Bel(\delta )}{\nu }\;log_{2}\;\frac{1-Bel(\delta )}{\nu }$$
where *ν =* 1 *+ PI(δ) − Bel(δ)*. With the value of *μ(Bel(δ), PI (δ))* approaching 0, the belief interval becomes larger, which leads to a lower confirmation of the hypothesis, so the formulation δ is considered more ambiguous.

#### 3.3.2. Semantic Representation of Object Task Affordance

Tasks are expressed as $S=\{\Theta_{S_{1}}\;\;,\Theta_{S_{2}}\;\;,\ldots ,\Theta_{S_{N}}\;\;\}$, which contains *N* different contextual task aspects. Each task aspect $S_{i}=\{\;\;\Theta_{S_{i,1}}\;,\Theta_{S_{i,2}}\;\;,\ldots ,\Theta_{S_{i,M}}\;\;\}$ contains a set of *M* mutually exclusive candidate high-level semantic task descriptions, serving as a BoE, and are denoted by $\left\{S_{i},m_{s_{i}}(\cdot )\right\}$. And,$m_{s_{i,j}}$ denotes the candidate’s quality value. Here, *i* ∈ {1, …, *N*} and *j* ∈{1, …, *M*}. The tasks and quality values for each aspect can be obtained from the established OT affordance.

We use nine binary contextual task aspects to characterize the features of tasks in real-world scenarios. Each task is represented with one positive feature and one negative feature, as shown in Table 3.

In Table 3, *Task(X, use, O_1_, O_2_)* indicates that the actuator *X* can use the tool object *O_1_* to perform the task *use* on *O_2_*. For example, a robot can use a hammer to perform the task of pounding a nail. When the user selects only one object, the expression automatically converts to *Task(X, use, O)*, which represents using object *O* to perform the task use without a passive recipient object. For instance, a robot can use a hammer to perform the task of pounding. Similarly, *Task(X, pourOut, O_1_, O_2_), Task(X, pourIn, O_1_, O_2_), Task(X, grasp, O), Task(X, press, O), Task(X, insertIn, O_1_, O_2_), Task(X, place, O_1_, O_2_), Task(X, push, O), and Task(X, pass, O)* follow the same rules, representing tasks such as *pourOut, pourIn, grasp, press, insertIn, place, push,* and *pass*, respectively.

#### 3.3.3. Semantic Representation of Object Action Affordance

The object affordance segmentation network serves as one of our underlying visual perception modules, capable of extracting the functional parts of objects. These functional parts are semantically described through their associated actions. Perception aspects are represented as $F=\{\Theta_{F_{1}}\;\;,\Theta_{F_{2}}\;\;,\ldots ,\Theta_{F_{N}}\;\;\}$, where *N* is the number of perception aspects. Each aspect $F_{i}=\{\;\;\Theta_{F_{i,1}}\;,\Theta_{F_{i,2}}\;\;,\ldots ,\Theta_{F_{i,M}}\;\;\}$ comprises a set of mutually exclusive high-level action semantics (a total of *M*), which serve as the Body of Evidence (BoE) and are denoted by $\left\{F_{i},m_{f_{i}}(\cdot )\right\}$. Their candidate quality values are denoted as $m_{F_{i,j}}$, where *i* ∈ {1, …, *N* } and *j* ∈ {1, …, *M*}. The actions and quality values of each aspect can be obtained from OA affordances.

Based on action affordances, we use ten binary perception aspects to represent the relevant actions. Each aspect has a positive side and a negative side, namely,$\Theta_{F_{i}}$ = {*f_i,_*_1_*, ¬ f_i,_*_1_}. The semantic descriptions of actions are shown in Table 4.

In Table 4, *grasp(X, O, part(O))* represents that robot *X* can perform the action *grasp* on a specific part of object *O*. For example, the robot can grasp the *holdingPart* of a cup. Similarly, actions such as *push(X, O, part(O)), press(X, O, part(O)), cutWith(X, O, part(O)), scoopWith(X, O, part(O)), pourWith(X, O, part(O)), insertInTo(X, O, part(O)), brushWith(X, O, part(O)), poundWith(X, O, part(O)),* and *placeOn(X, O, part(O))* indicate that robot *X* can, respectively, perform the actions *push, press, cutWith, scoopWith, pourWith, insertInTo, brushWith, poundWith,* and *placeOn* on a specific part of object O.

#### 3.3.4. Semantic Representation of Action Sequence

When a robot performs a task, it must clearly understand how to interact with an object. In this section, we will discuss the semantic representation of inferred action sequences.

The semantic representation of an action sequence is given as $A=\{\Theta_{A_{1}}\;\;,\ldots ,\Theta_{A_{N}}\;\;\}$, which consists of *N* distinct action sequences. Each aspect $A_{i}=\{\;\;\Theta_{A_{i,1}}\;,\Theta_{A_{i,2}}\;\;,\ldots ,\Theta_{A_{i,M}}\;\;\}$ contains a set of M mutually exclusive high-level semantic descriptions that serve as a BoE and are denoted by $\left\{\Theta_{A_{i}},m_{A_{i}}(\cdot )\right\}$. The candidate quality values are represented by $m_{A_{i\;,j}}$, where i∈{1,…,N} and j∈{1,…,M}. The action sequence affordances and quality values of each aspect can be derived from our semantic constraint rule models.

We use eleven binary affordance aspects to represent the likelihood between the robot and the action sequence being executed. Each aspect consists of a positive side and a negative side, denoted as $\Theta_{A_{i}\;}\;=\{a_{i,1}\;,\neg a_{i,1}\;\}$. The semantic descriptions of the action sequences are shown in Table 5.

Table 5 presents the semantic representation of action sequences, encoding robot–user object interactions in a structured format. Each entry, such as “*X grasp part(O) pourWith part(O)*”, indicates that the robot (*X*) first grasps a specific part of object *O* (e.g., the handle of a mug) and then performs the “*pourWith*” action using the same or another part (e.g., the mug’s spout). Similarly, “*X grasp part(O_1_) placeOn part(O_2_)*” denotes grasping a part of object *O_1_* (e.g., a knife’s handle) and placing it on a part of object *O_2_* (e.g., a table’s surface). This notation leverages object affordances to map user intents to executable steps. For instance, in a kitchen scenario, “*X grasp part(mug) pourWith part(mug)*” signifies grasping the mug’s handle and pouring from its spout, ensuring clarity and precision.

#### 3.3.5. Semantic Constraint Rules Model for Fusion of OT and OA Affordance

We use D-S theory to represent object *O* and its functional part *P*, as well as the user’s possible task intention *T*, action intention *S*, and action sequence intention *A*. To impose semantic constraints on the robot’s reasoning of action sequences, we combine these elements to construct a semantic constraint rule set *R*. The semantic representation of the rule set is $R=\{\Theta_{R_{1}},\ldots ,\Theta_{R_{N}}\}$, which includes *N* different rule aspects. Each rule $\Theta_{R_{i}}$ contains *M* mutually exclusive candidate high-level semantic description rules, which serve as a Body of Evidence (BoE) and are described by $\left\{\Theta_{R_{i}},m_{R_{i}}(\cdot )\right\}$. Here,$m_{r_{i\;,j}}$ represents the perceived candidate quality value, where i∈{1,…,N} and j∈{1,…,M}. Each aspect has two directions: positive and negative, thus $\Theta_{R_{i}\;}\;=\{r_{i,1}\;,\neg r_{i,1}\;\}$.

For the object *O* that the user focuses on and their functional parts *P*, we can use D-S theory to integrate object task and object action affordance to achieve more accurate reasoning of the action sequence intention. The semantic constraint rule model for users’ intention reasoning is expressed as $r_{m_{o,p\to f,s\to a}}^{i,j}:=o\wedge p\Rightarrow f\wedge s\Rightarrow a\;$.

Given a BoE $\left\{\Theta_{R_{i}},m_{R_{i}}(\cdot )\right\}$, the belief function *Bel(R) = α* is used to calculate the belief of the rule *R*. The plausibility function of *R* is *PI(R) = β*. Thus, the belief interval of the rule *R* is *[α, β]*, which can be used to replace $m_{o,p\to f,s\to a}$. The semantic constraint rule model can be changed to the form $r_{\left[\alpha_{i,j},\;\beta_{i,j}\right]}^{i,j}:=o\wedge p\Rightarrow f\wedge s\Rightarrow a\;$.

Based on the logical reasoning model, we defined fusion rules for user action intention sequence reasoning, some of which are as follows:
$\begin{array}{l} r_{\left[0.8,1\right]}^{1}\;:=Object(O)\wedge holdingPart(O)\wedge poundingPart(O)\\ \Rightarrow Task\left(X,\;use,\;O\right)\wedge grasp\left(X,\;O,\;holdingPart\left(O\right)\right)\wedge poundWith\left(X,\;O,poundingPart\left(O\right)\right)\;\\ \Rightarrow X\;grasp\;holdingPart(O)\;poundWith\;poundingPart(O) \end{array}$$\begin{array}{l} r_{\left[0.8,1\right]}^{6}\;:=Object(O)\wedge poundingPart(O)\\ \Rightarrow Task\left(X,pass,O\right)\wedge grasp\left(X,\;O,\;poundingPart(O)\right)\\ \Rightarrow X\;grasp\;poundingPart(O) \end{array}$$\begin{array}{l} r_{\left[0.8,1\right]}^{12}\;:=Object(O)\wedge holdingPart(O)\\ \Rightarrow Task\left(X,grasp,O\right)\wedge grasp\left(X,\;O,\;holdingPart\left(O\right)\right)\\ \Rightarrow X\;grasp\;holdingPart(O) \end{array}$

When the object consists of two parts, *holdingPart(O)* and *poundingPart(O)*, the inference rules *r^1^*, *r^6^*, and *r^12^* may be applicable. Rule *r^1^* represents that the robot grasps the *holdingPart* of object *O* and uses the *poundingPart* of object *O* to perform the *use* task. Rule *r^6^* represents the robot grasping the *poundingPart* to perform the *pass* task. Rule *r^12^* represents the robot grasping the *holdingPart* to perform the *grasp* task. And then, using D-S theory, the belief function and plausibility function are calculated to obtain the confidence interval [*Bel(A), PI(A)*]. Based on Formula (3), the confidence value is calculated, and recommendations are made to the user according to the size of the confidence value. The calculation process using the rules will be demonstrated in the next section.

#### 3.3.6. Action Sequence Intention Inference Under OT and OA Affordance Constraints

The reasoning process uses low-level perceptual information as input parameters. The object recognition network is employed to obtain the category and name of the object, while the affordance segmentation network extracts the functional parts of the object and probabilistically maps the visual features of the functional parts to corresponding actions. Subsequently, this information is encoded using D-S theory. And, the object/task ontology is encoded into uncertain logical rules through CP-Logic and further integrated into task-related evidence using D-S theory. Finally, based on D-S theory, the uncertainty interval of the user action sequence intent can be derived, where ⊙ and ⊗ represent “modus ponens” and “And” in D-S theory, respectively. For the derived uncertainty interval, Formula (3) is used to determine the threshold of the action sequence intent. By comparing the threshold values, the system recommends potential action options to the user to confirm their true intent.

Figure 4 illustrates the calculation processes of D-S theory fusion for OT (object task) and OA (object action) affordances. Taking the object “knife” as an example, suppose the user focuses on the knife during a specific interaction. The robot first detects the knife and identifies its functional components. Through object task affordances, the robot determines applicable tasks for the knife, such as *use*, *grasp*, and *pass*, along with their respective probabilities. Simultaneously, through object action affordances, the robot identifies actions suitable for different functional parts of the knife, such as *cutWith(X, O, cuttingPart(O))*, *grasp(X, O, cuttingPart(O))*, and *grasp(X, O, holdingPart(O))*, along with their associated probabilities. Finally, based on the fusion rules of D-S theory (e.g., *r^2^, r^12^, r^7^, r^14^*), the robot computes the belief function and the belief interval to obtain the belief interval [*Bel(A), PI(A)*]. Using Equation (3), the robot calculates the confidence value. The robot will first recommend the action sequence intention with the highest confidence value (*X grasp holdingPart(knife) cutWith cuttingPart(knife)*) to the user and then sequentially suggest action sequence intentions with slightly lower confidence values until the user’s true intention is ultimately confirmed.

## 4. Experiments and Results

The proposed method is developed on the WMRA, which is equipped with an embedded NVIDIA Jetson TX2 board. As shown in Figure 5, the WMRA consists of components such as a robotic arm and an electric wheelchair and is equipped with Intel Realsense D435i and Intel Realsense D435 RGB-D cameras. The robot uses a visual system to capture environment information and uses Kinova Jaco GEN2 6-DOF 3-Finger Arm to manipulate objects based on the ROS framework.

We conducted experiments on the training of the model (we call it the User Habit Adaptation Experiment) and the inference of intentions after the completion of training for each of the three parts of the proposed method. The parameter values are set as follows: *α* = 0.1, *w_i_* = 1/(*i* + 1) and *i* = 1, 2,…, *n* (*n* is the total number of feedbacks).

### 4.1. Object Recognition and Segmentation of Object Functional Regions Experiment

We developed a customized object detection dataset specifically designed for household scenarios and trained it using the YOLOv8 algorithm. The dataset was processed with random brightness enhancement, random rotation, and salt and pepper noise to adapt to diverse household objects and environments, ensuring robustness and generalization. Additionally, we deployed the YOLOv8 algorithm on the embedded Jetson TX2 development board, achieving efficient and accurate real-time object detection. The detection results of some objects are shown in Figure 6.

Furthermore, in our experiments, we used an RGB-D camera to capture color and depth images. In order to perform instance segmentation of objects, we applied the classic Mask R-CNN instance segmentation algorithm. We built a dataset containing 3600 images, trained it for 100 epochs on an NVIDIA GeForce RTX 4060, and successfully deployed the algorithm integrated into ROS. The segmentation results are shown in Figure 6.

### 4.2. Task Intentions Reasoning Experiment

#### 4.2.1. User Habit Adaptation Experiment

Evaluation of User Initial Habit-Learning Ability for Single Object

This experiment aims to evaluate our task intention inference model’s ability to learn and adapt to the first users’ habits from initialization. The experimental data come from the real-life usage records of users interacting with objects, which we refer to as “habits”. During the experiment, we selected one object from each of the four categories, involving nine tasks, as shown in Table 6.

For the selected four objects, we observed and recorded the tasks performed by three participants (Subject #1, Subject #2, and Subject #3) in their daily lives. Due to certain limitations, we selected other colleagues from the research institute as experimenters, including both male and female participants. Each object required 110 operation habit records. Table 7 shows some of the operation records for the mug object. Based on the data in Table 7, there are seven possible tasks for the mug object. Our rule is that when the participant focuses on the mug and performs a task, the task is recorded as 1 and the other tasks are recorded as 0. For example, in the first record, if the participant performs the task “*pass*” with the mug, we mark the “*pass*” task as 1 and the other six tasks (*pourOut, grasp, place, push, pourIn, and insertIn*) as 0.

During the initial habit-learning training process, we used the life records of Subject #1 as the study sample, taking the mug as an example. The first 100 records were used as training data while the last 10 records were used as testing data. In the first round of training, we utilized the first data entry from Table 8 (representing one time-step of the life record) for training. The initial probability of the task intention inference model was set to 0.1. When all probabilities were equal to 0.1 or identical, the inference model made its first prediction by randomly recommending an object.

Once the model provides the inference results, we obtain user feedback (affirming or denying the inference results) through interaction until the user affirms the inference results. At this point, the current round of training ends, and the model is updated based on the user’s confirmed recommendation results. After completing this round of training, the next round of training begins, using the second data entry from Table 7. At each time-step of the life record, we recorded the changes in the probability of each item being suitable for different tasks, as shown in Table 7.

The results of the entire training process are shown in Figure 7a,d,g,j, which detail the model’s performance at different stages and specifically illustrate the trend of the probability curves of users performing tasks with items over time-steps. As can be seen from the figures, during the first 30 time-steps of training, the model’s learning process exhibits significant dynamic characteristics: for tasks frequently performed by the user, the corresponding probability curves show a gradual upward trend, while for tasks less frequently performed by the user, the probability curves gradually decline. This phenomenon clearly indicates that the task intention inference model is gradually learning and adapting to the user’s operational habits with items through continuous training.

After 30 time-steps, the situation changes: the probability curves of items and tasks begin to stabilize and no longer show significant fluctuations. This turning point indicates that the model, after the initial learning and adjustment, has successfully adapted to the user’s task habits with items to a certain extent and has reached a relatively convergent state. At this point, the model is not only able to accurately identify the user’s operational patterns but also maintain consistency and reliability in subsequent task intention inference.

Evaluation of User Habit Switching Learning Ability for Single Object

This experiment aims to evaluate the model’s ability to relearn and adapt to another user’s object operation habits after having already adapted to one user’s habits. The objects and tasks involved in the experiment remain consistent with those in the previous section, and the training data are derived from Subject #2 and Subject #3. The experimental results are shown in Figure 7.

The three columns of probability curves in Figure 7 illustrate the model’s learning and adaptation processes under different user habits. The left column reflects the task intent inference model’s process of learning and adapting to Subject #1’s task habits. The middle column demonstrates the model’s subsequent relearning and adaptation to Subject #2’s task habits after mastering Subject #1’s lifestyle patterns. The right column depicts the model’s further adjustment and adaptation to Subject #3’s task habits following its adaptation to Subject #2’s lifestyle patterns.

By comparing the middle column of Figure 7 with the left column, it can be observed that within the first 30 time-steps of the middle column, the model’s probability curves exhibit significant fluctuations. This indicates that the model is transitioning from adapting to Subject #1’s habits to rapidly learning and shifting toward Subject #2’s operational habits. During this initial phase, the model adjusts itself by distinguishing the frequency of different tasks. For tasks frequently performed by the user—such as *place-remote, place-mug, pass-mug, place-knife, and pass-knife*—the model assigns progressively higher probabilities, as indicated by the steadily increasing trends in the probability curves. Conversely, for tasks rarely performed—such as *press-remote, grasp-remote, pourOut-mug, grasp-mug, use-knife, and grasp-knife*—the probabilities gradually decrease, as reflected by the downward trends in the corresponding curves.

After 30 time-steps, the probability curves stabilize and no longer exhibit significant fluctuations. This suggests that the model has largely mastered Subject #2’s operational habits, reaching a relatively convergent state. At this point, the model’s predictions for subsequent task intents become more stable and reliable. Once it identifies the user’s focus or intent, it can maintain consistency and accuracy in its follow-up predictions.

A similar phenomenon and outcome can be observed when comparing the right column of Figure 7 with the middle column. The model initially learns and distinguishes task frequencies for the new user (Subject #3), after which the probability curves gradually converge, indicating that the model has effectively adapted to Subject #3’s operational habits.

From the three figures in the first row, it is evident that the probability curves for the object ‘chair’ and its associated tasks remain relatively stable across the right, middle, and left figures. This indicates that the intent inference model does not require significant relearning or adaptation to another user’s habits, suggesting a degree of similarity in the participants’ operational habits for the chair. This further validates that our action intent inference method can effectively learn and adapt to user habits during transitions between users, irrespective of the extent of differences in their habits.

Evaluation of User Initial Habit-Learning Ability for Multiple Objects

We also evaluated the model’s reasoning ability when users focus on multiple objects, beginning with an analysis of its process from initialization to learning and adapting to Subject #1’s habits. The objects and tasks of interest are listed in Table 8: one group comprises the knife and table with their associated tasks “*use*” and “*place*” while the other includes the bottle and mug with their respective tasks *“insertIn”, “pourOut”, and “pourIn”.* Following the same methodology as in the previous section, we observed and recorded the tasks performed by two experimenters on these objects during daily activities, with results summarized in Table 8.

During the initial habit-learning training process for multiple objects, we utilized Subject #1’s daily activity records. The results of the entire training process are presented in Figure 8a,d. As illustrated, within the first 30 time-steps, the model’s probability curves exhibit significant fluctuations. For tasks frequently performed by the user, the corresponding probability curves show a gradually increasing trend, whereas for tasks rarely performed, the curves display a downward trend. This phenomenon clearly indicates that the task intent inference model is progressively learning and adapting to the user’s object-handling habits through continuous training. After 30 time-steps, the probability curves associated with multiple objects stabilize, indicating that the model has effectively learned the user’s daily habits, achieved a relatively convergent state, and enabled relatively accurate habit predictions.

Evaluation of User Habit Switching Learning Ability for Multiple Objects

This experiment aims to evaluate the model’s ability to relearn and adapt to another user’s object operation habits after having already adapted to one user’s habits. The objects and tasks involved in the experiment remain consistent with those in the previous section, and the training data are derived from Subject #2 and Subject #3. The experimental results are shown in Figure 8.

The three columns of probability curves in Figure 8 illustrate the model’s learning and adaptation processes under different user habits. The left column reflects the task intent inference model’s process of learning and adapting to Subject #1’s task habits. The middle column demonstrates the model’s subsequent relearning and adaptation to Subject #2’s task habits after mastering Subject #1’s lifestyle patterns. The right column depicts the model’s further adjustment and adaptation to Subject #3’s task habits following its adaptation to Subject #2’s lifestyle patterns.

From these three columns of figures, it can be observed that the probability curves for objects and their corresponding tasks remain relatively stable across the right, middle, and left columns. This suggests that the intent inference model does not undergo a pronounced process of relearning and adapting to another user’s habits, indicating a degree of similarity in task habits for multiple objects among the participants. This also demonstrates that our action intent inference method can effectively learn and adapt to user habits when transitioning between users, regardless of whether significant differences exist between their habits.

#### 4.2.2. Task Intention Inference

As time progresses, the model gradually learns and adapts to the user’s daily habits. To evaluate changes in the model’s intent prediction capability during the training process, we selected the model’s training states at the 5th, 15th, and 50th time-steps for intent prediction. The target intents to be predicted were derived from the last 10 habit records of Subject #1 (a total of 110 operational habit records per object, with the first 100 used for model training).

During each reasoning process (i.e., within a single time-step), we recorded the number of interactions required between the reasoning model and the user until the task intent was accurately predicted. After completing the prediction of 10 habit records for Subject #1, considering the variability in user interaction and model prediction, we repeated the prediction process five times. Finally, we calculated the average number of user interactions and their standard deviations over 200 intent prediction processes (across five rounds for four objects) and plotted the results, as shown in Figure 9. A lower number of interactions in a single reasoning process indicates a stronger prediction capability. For example, if the recorded number of interactions is one, it signifies that the reasoning model accurately predicted the user’s task intent on the first interaction.

The prediction results are presented in Figure 9. As shown, with an increase in time-steps, the number of interactions required between the task intent inference model and the user for accurate predictions decreases. This indicates that the model progressively learns and adapts to the user’s habits, with its accuracy in reasoning task intents improving over time, consistent with the analysis in previous sections.

#### 4.2.3. Effect of Learning Rates on the Model’s Performance in Learning User Habits

We evaluated the role of the learning rate parameter in the habit adaptation process. As described in Section 3.1.4, we employed reinforcement learning to update the user’s habits, with the learning rate parameter used to control the weight between current and historical feedback. Using a series of learning rates (α = 0.01, 0.05, 0.1, 0.4, 0.7), we conducted experiments on both single-object and multi-object user habit learning. These experiments focused on a specific set of object task intents to investigate the impact of different learning rates on model performance.

The experimental results are presented in Figure 10. As clearly observed from the figure, the learning rate significantly influences the processes of habit adaptation and habit-switching learning. Specifically, when the learning rate is set to a low level, such as α = 0.01 or α = 0.05, the model’s adaptation speed is notably slow. At these values, the probability curves exhibit a gradual upward trend, indicating that the model requires more time-steps to incrementally accumulate information about the user’s habits and achieve an accurate understanding and prediction of their behavior. This slower adaptation process suggests that a low learning rate imparts a degree of conservatism to the model during learning, potentially limiting its ability to fully leverage current feedback and thus prolonging the habit adaptation cycle.

In contrast, when the learning rate is adjusted to a moderate level, such as α = 0.1, the model demonstrates a more desirable adaptation capability. Under this condition, the probability curves show a stable and consistent growth trend, eventually stabilizing at the correct task intent. This behavior indicates that the model can effectively learn the user’s habits within a reasonable timeframe, avoiding delays due to excessively slow learning while also preventing uncertainty caused by overly rapid adjustments. Experimental data further reveal that a learning rate of α = 0.1 enables the model to achieve optimal performance in both habit adaptation and habit-switching learning, striking an effective balance between learning speed and prediction accuracy. All other experiments we conducted were based on a learning rate of α = 0.1.

However, when the learning rate is increased to higher levels, such as α = 0.4 or α = 0.7, the dynamics shift. Although the initial adaptation speed accelerates, the probability curves begin to exhibit noticeable fluctuations or instability. This instability may stem from an excessively high learning rate causing the model to over-rely on current feedback data while neglecting the cumulative effect of historical information, potentially leading to risks of overfitting or behavioral inconsistencies during learning. Such fluctuations not only undermine the model’s ability to accurately learn user habits but may also reduce the reliability of its predictions in practical applications, posing potential negative impacts on the user experience.

### 4.3. Action Intentions Inference Experiment

Our ultimate goal is to enable the robot to accurately identify the user’s task intention and reliably perform corresponding actions on various parts of the object to complete the task. Therefore, when reasoning about the user’s task intention regarding an object, the robot needs to not only determine the type of task but also to understand the actions that can be performed on different parts of the object, thereby identifying the specific operation the user requires. To validate this capability, we previously configured a task intention prediction experiment and further configured an action intention prediction experiment in this section. As shown in Table 9, the task intention “*place-chair*” corresponds to the action intention “*placeOn supportingPart*” and so on. In other words, when the task intention is “*place-chair*”, the predicted action intention should be “*placeOn supportingPart*”. This experimental setup ensures consistency between task intentions and action intentions. Additionally, we record the number of interactions with the user required until the prediction is accurate. The experimental setup and data processing are consistent with those in Section 4.2.2.

The initial probability assignment in the action intention inference model considers only the geometric features and functional constraints of an object’s functional components, without taking into account the user’s habits or preferences. Therefore, we only evaluated the model’s action intention prediction capability. In the experiment, we used only the last 10 habitual records of Subject #1 for action intention prediction. For a single object, the action intention inference results are shown in Figure 11a.

As shown in the figure, the average number of interactions during the action intention inference process is approximately 3.5, significantly higher than the average number of interactions in the task intention inference section. This may be because the inference process focuses solely on a single functional part of the object without considering the connections between different functional parts.

For multiple objects, the action intention reasoning results are shown in Figure 11b. As shown in Figure 11b, during action intention inference, the average number of interactions for multiple objects intent reasoning is approximately five, which is significantly higher than that of task intention inference. This discrepancy may arise because action intention inference for multiple objects primarily involves combining the actions of different object parts. Without the contextual constraints of a specific task, it becomes more challenging to accurately determine the user’s intended actions.

### 4.4. Action Sequences Intention Inference Experiment

Based on D-S theory, we perform action sequence intention inference under multi-dimensional constraints (including the task intention constraints of the object and the action intention constraints of the functional parts of the object). This enables the robot to accurately infer the user’s task intention and action sequence intention, reasonably select the functional parts of the object, and execute corresponding actions during the task execution process, thereby reliably assisting the user in completing household tasks and reducing the burden of operating the robotic arm.

To quantitatively evaluate the proposed method, we conducted a series of experiments, including random action sequence intention prediction experiments and ablation experiments. The core objectives of the experiments are, first, to determine the number of interactions with the user required for the model to predict a specific action sequence intention, and second, to evaluate the advantages of our method in intention prediction. When the user focuses on the knife and intends to use it for cutting, the model should be able to accurately predict the action sequence: *X grasp holdingPart(Knife) cutWith cuttingPart(knife)*. We will record the number of interactions required with the user until the prediction is accurate.

#### 4.4.1. Action Sequence Intention Inference

In previous sections, we designed task intention and action intention prediction experiments. Similarly, in this section, we conducted action sequence intention prediction experiments. For a single object, the experimental configuration is shown in Table 10. For multiple objects, we similarly conducted intention prediction experiments, with specific settings shown in Table 11. The experimental setup and data processing are consistent with those in Section 4.2.2.

For example, the task intention *place-chair* corresponds to the action sequence intention reasoning rule *r^29^*. In other words, when the task intention is *place-chair*, the action sequence intention to be predicted is *r^29^*, represented as *X placeOn supportingPart(O)*. We recorded the number of interactions required for the model to accurately predict this intention.

As shown in Table 11, the task intention *Task(X, place, knife, table)* corresponds to the action sequence intention reasoning rule *r^28^* for multiple objects.

For a single object, the action sequence intention reasoning results are shown in Figure 12a. For multiple objects, the action sequence intention reasoning results are shown in Figure 12b.

#### 4.4.2. Ablation Experiments

To thoroughly validate the effectiveness of our proposed method, we designed and conducted ablation experiments integrating various modules to systematically demonstrate substantial improvements in model performance. Specifically, we assessed the contributions of individual key modules to overall model performance through separation and combination experiments. These modules include: (1) the object action (OA) module; (2) the object task (OT) module; and (3) the D-S module. To ensure the scientific rigor and reproducibility of the experiments, we meticulously recorded the number of interactions required between the model and the user to accurately predict the intent for each object. Additionally, we repeated the intent prediction experiment for five rounds. Subsequently, we conducted a statistical analysis of all the collected data, computing the mean number of interactions and their standard deviations to quantify performance variations across different configurations. The experimental setup and data processing are consistent with those in Section 4.2.2. And, the experimental results are comprehensively presented in Table 12.

To further validate the performance advantages of our approach, we conducted in-depth ablation experiments analyzing both single-object and multi-object scenarios. For the single-object case, the results of the ablation experiments are presented in Figure 13a. A comprehensive analysis of Figure 13a and Table 12 clearly reveals that, compared to the task intention reasoning method (E2: OT), our proposed method (E3: Ours) significantly optimizes the average number of interactions, achieving a reduction of 14.085%. Furthermore, compared to the action intention reasoning method (E1: OA), our method (E3: Ours) achieves a substantial reduction in the average number of interactions by 52.713%. The experimental results demonstrate that, by more precisely reasoning the user’s true intentions, our method not only enhances prediction accuracy but also substantially minimizes unnecessary user interactions. This efficiency is particularly pronounced in the single-object scenario.

For the multi-object scenario, the results of the ablation experiments are presented in Figure 13b and Table 12. As shown in the figure, in a multi-object environment, both the task intention reasoning method (E2: OT) and our proposed method (E3: Ours) significantly reduce the average number of interactions compared to the action intention reasoning method (E1: OA), achieving a reduction of 71.477%. However, our method (E3: Ours) exhibits a performance consistent with the task intention reasoning method (E2: OT) in terms of average interaction counts. This is because, in multi-object intent reasoning, the constraints imposed by inter-object relationships and the number of objects significantly limit the scope of inference, resulting in fewer possible intent outcomes (typically one or two). For instance, when involving a “knife” and a “table”, user intents may be confined to “*Task(X, use, knife, table)*” or “*Task(X, place, knife, table)*” This limitation makes it challenging for the model to reduce the number of interactions through additional modules during the inference process. Both E2 (OT) and E3 (Ours) effectively capture these intents through task-level reasoning, keeping consistent interaction counts. Equally important, compared to E2 (OT), E3 (Ours) integrates the OA module, leveraging object affordances to specify functional parts of objects in task execution (e.g., the “*cuttingPart* of the knife” for “cutting”), thereby ensuring reliable task performance.

## 5. Conclusions

In this paper, we propose an innovative intention reasoning method for users’ action sequences by fusing object task and object action affordances based on D-S theory. This method combines the advantages of probabilistic reasoning and visual affordance detection to establish an affordance model for objects and potential tasks or actions based on user usage habits and object attributes. This facilitates encoding object task affordance and object action affordance using D-S theory to perform action sequence reasoning. By leveraging this algorithm, the user can convey operational intentions to the robot through “gazing at” or “selecting” objects and reliably perform corresponding actions on different parts of the object to complete the task, significantly reducing the physical burden of controlling the WMRA.

First, deep learning techniques such as YOLOv8 and Mask R-CNN are utilized to acquire visual information, including object categories and the segmentation of functional regions. Subsequently, we constructed an ontology-based knowledge model for commonly used objects and tasks in household environments to explicitly define the object task affordance relationships. These relationships were probabilistically encoded using CP-Logic, thus establishing the task reasoning module. This module can learn and adapt to users’ historical operational habits, enabling more accurate reasoning of the implicit task intentions associated with the objects of interest. The experimental results demonstrate that our model can effectively switch and adapt to different user habits, and when the learning rate parameter α is set to 0.1, a desirable balance between learning efficiency and reasoning accuracy can be achieved.

The action reasoning module in this work is based on Mask R-CNN, which detects visual affordance regions of objects. These affordance regions are then mapped to actions by integrating the objects’ geometric features and functional constraints, establishing the object action affordance reasoning model. However, the initial probability assignment in the action intention inference model considers only the geometric features and functional constraints of an object’s functional components, without taking into account the user’s habits or preferences. Finally, this study incorporates D-S theory to encode and fuse information from the aforementioned reasoning modules, thereby inferring the action sequence intention for the target object. The generated action sequence guides the robot in object manipulation under the constraints of tasks, actions, and objects. Through this algorithm, the WMRA can not only accurately infer the user’s intentions but also execute appropriate actions on the functional regions of objects during real-world operations to ensure reliable task execution. As a result, it effectively assists users in completing household tasks and reduces the physical burden on disabled users when controlling the WMRA.

Experimental evaluations further highlight the method’s strengths. For single-object scenarios, our approach (E3: Ours) reduces the average number of interactions by 14.085% compared to the task intention reasoning method (E2: OT) and by 52.713% compared to the action intention reasoning method (E1: OA), achieving an average of 1.35 interactions. In multi-object scenarios, both our method (E3: Ours) and the task intention reasoning method (E2: OT) outperform the action intention reasoning method (E1: OA), achieving a substantial reduction of 71.477% in average interaction counts. However, due to constraints imposed by inter-object relationships and the limited number of possible intent outcomes, no significant difference is observed between E3 (Ours) and E2 (OT) in terms of average interaction counts. Nevertheless, our method (E3: Ours) enhances task execution reliability by integrating the OA module, which leverages affordances to specify functional parts of objects, thereby ensuring system robustness.

Overall, our intention reasoning method significantly enhances the WMRA’s ability to understand users’ intents. The reduction in interaction counts and the adaptability to varying user habits affirm its user-friendliness and practical utility. Future work could explore incorporating real-time user feedback and expanding the ontology to include a broader range of objects and tasks, further improving the method’s scalability and applicability in diverse assistive scenarios.

## Figures and Tables

**Figure 1 sensors-25-01992-f001:**
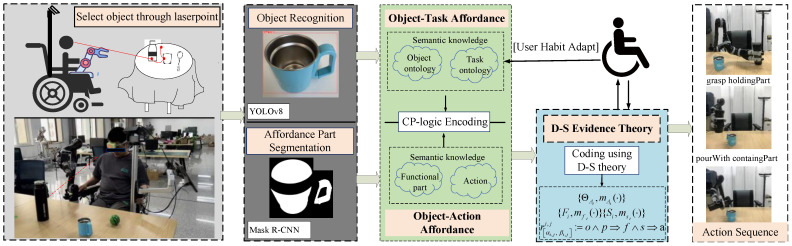
Intent reasoning framework of the WMRA.

**Figure 2 sensors-25-01992-f002:**
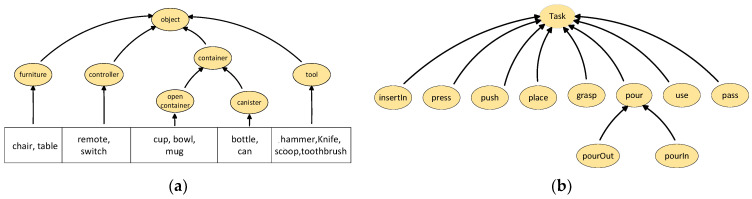
Ontology description: (**a**) object ontology; (**b**) task ontology.

**Figure 3 sensors-25-01992-f003:**
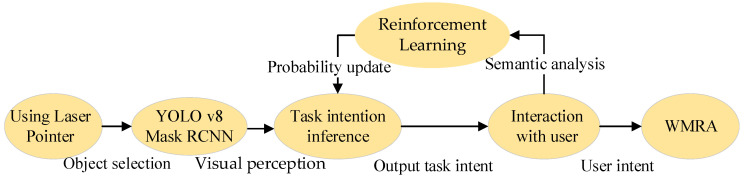
User habit adaptation framework.

**Figure 4 sensors-25-01992-f004:**
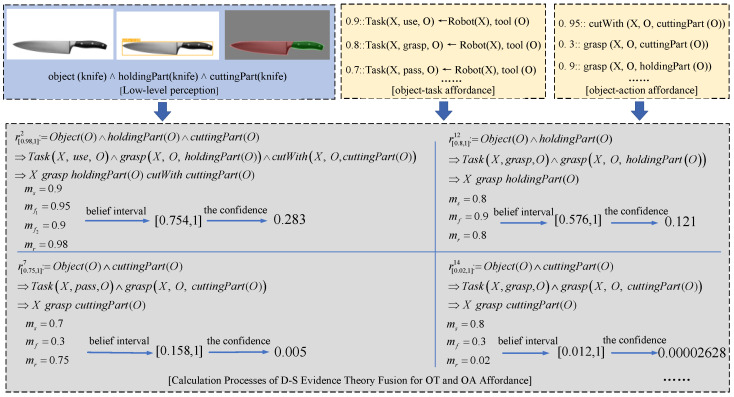
Calculation processes of D-S theory fusion for OT and OA affordance.

**Figure 5 sensors-25-01992-f005:**
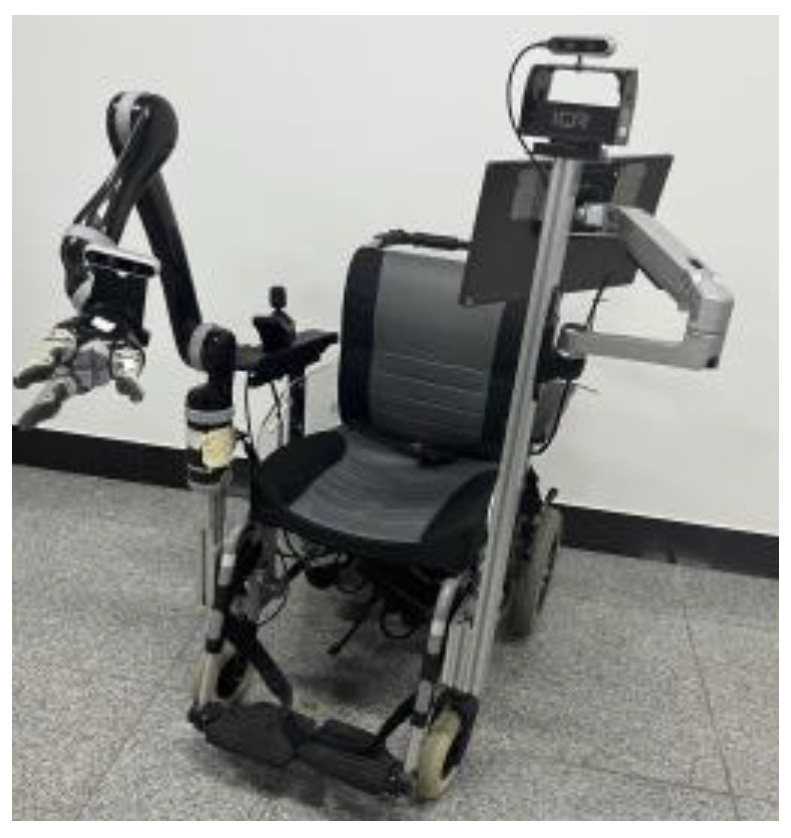
Experimental platform.

**Figure 6 sensors-25-01992-f006:**
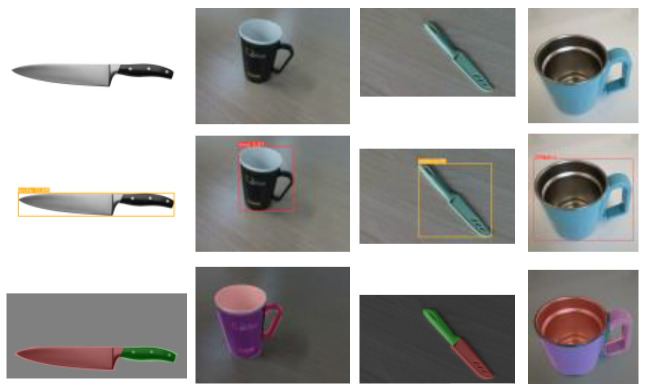
Examples of object recognition and object functional region segmentation.

**Figure 7 sensors-25-01992-f007:**
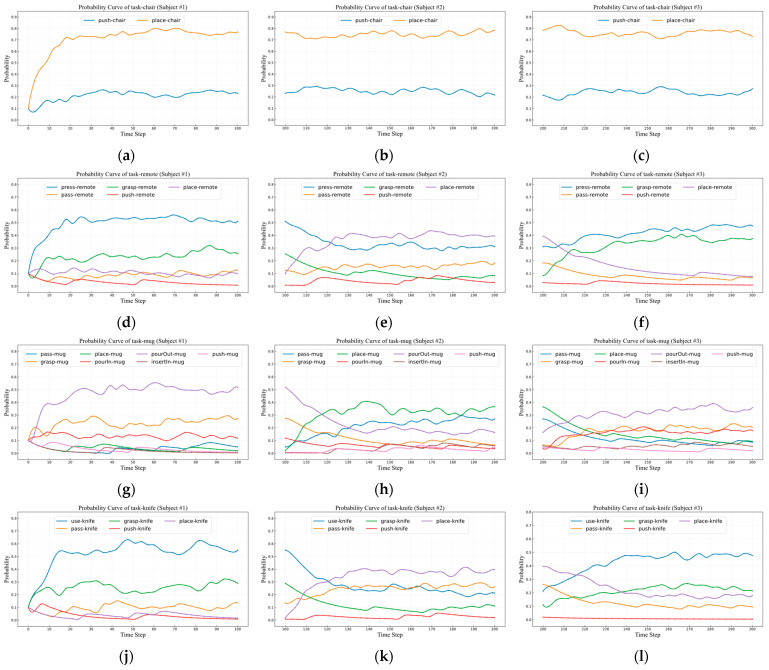
Probability curves of task object throughout the training process for single object: (**a**,**d**,**g**,**j**) show curves for chair, remote, mug, and knife with Subject #1; (**b**,**e**,**h**,**k**) with Subject #2; and (**c**,**f**,**i**,**l**) with Subject #3.

**Figure 8 sensors-25-01992-f008:**
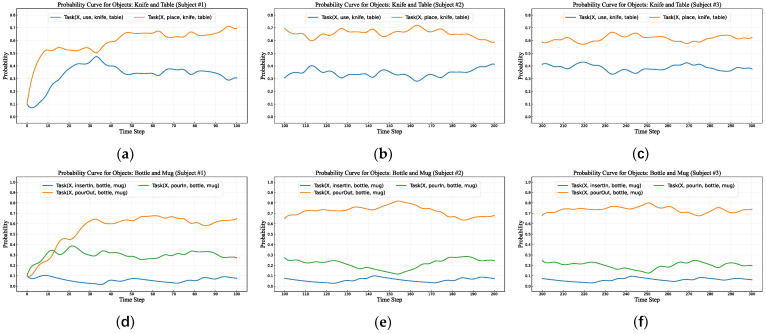
Probability curves of task and object during training for multiple objects: (**a**) shows curves for knife and table with Subject #1; (**d**) for bottle and mug with Subject #1; (**b**) for knife and table with Subject #2; (**e**) for bottle and mug with Subject #2; (**c**) for knife and table with Subject #3; (**f**) for bottle and mug with Subject #3.

**Figure 9 sensors-25-01992-f009:**
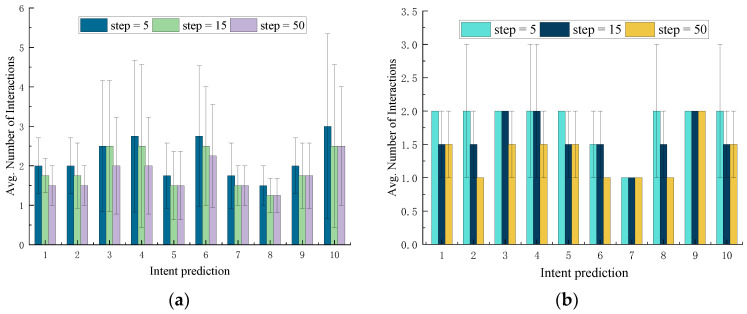
Model behavior at time-step 5, 15, and 50 for intent prediction. The error bars represent the standard deviation in the number of interactions. (**a**) Intent prediction results for single object during the initial habit-learning process at different time-steps; (**b**) intent prediction results for multiple objects during the initial habit-learning process at different time-steps.

**Figure 10 sensors-25-01992-f010:**
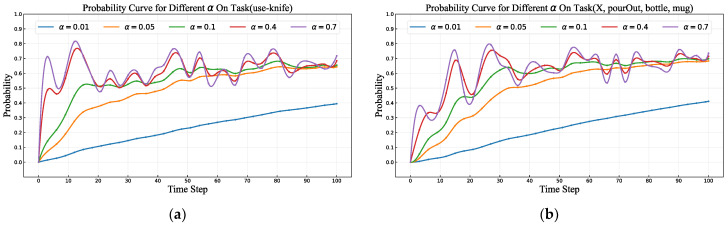
Effect of varying learning rates (α) on the model’s user habit-learning performance. (**a**) Probability curves for *task(use-knife)* during initial habit adaptation for single object under different learning rates. (**b**) Probability curves for *task(use-knife)* during habit-switching learning for single object under different learning rates.

**Figure 11 sensors-25-01992-f011:**
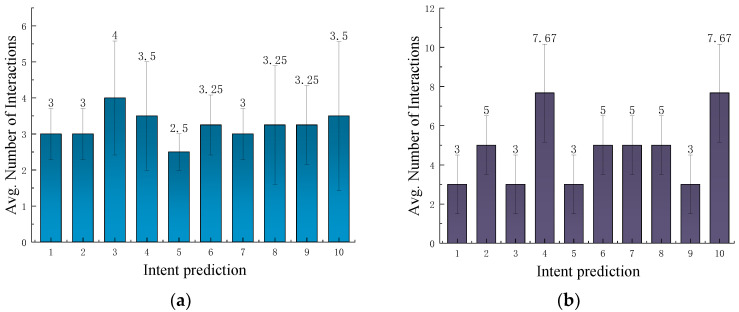
User action intent prediction results. The error bars represent the standard deviation in the number of interactions. (**a**) User action intent prediction results for single object. (**b**) User action intent prediction results for multiple objects.

**Figure 12 sensors-25-01992-f012:**
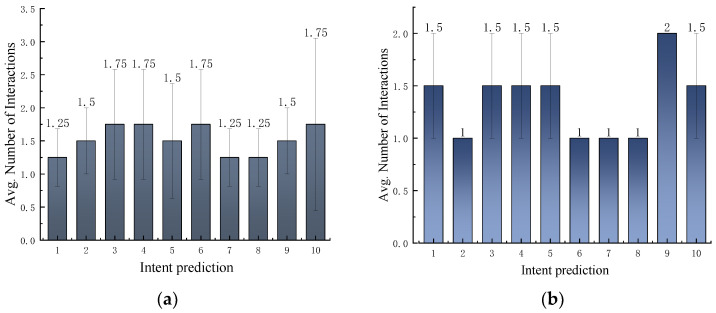
User action sequence intent prediction results. The error bars represent the standard deviation in the number of interactions. (**a**) User action sequence intent prediction results for single objects. (**b**) User action sequence intent prediction results for multiple objects.

**Figure 13 sensors-25-01992-f013:**
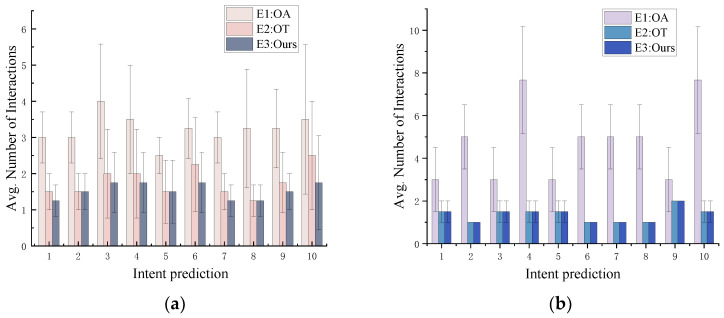
The results of ablation experiments. The error bars represent the standard deviation in the number of interactions. (**a**) The results of ablation experiments for single object. (**b**) The results of ablation experiments for multiple objects.

**Table 1 sensors-25-01992-t001:** Object task affordance description. Symbol “✓” represents that the task has some relations with the object.

Object Task Affordance		Furniture		Controller		Container					Tool			
						Open Container			Canister					
		Chair	Table	Remote	Switch	Cup	Bowl	Mug	Bottle	Can	Knife	Scoop	Toothbrush	Hammer
*pass*				✓	✓	✓	✓	✓	✓	✓	✓	✓	✓	✓
*use*											✓	✓	✓	✓
*pour*	*In*					✓	✓	✓						
	*Out*					✓	✓	✓	✓	✓				
*grasp*				✓	✓	✓	✓	✓	✓	✓	✓	✓	✓	✓
*place*		✓	✓	✓	✓	✓	✓	✓	✓	✓	✓	✓	✓	✓
*push*		✓	✓	✓	✓	✓	✓	✓	✓	✓	✓	✓	✓	✓
*press*				✓	✓									
*insertIn*						✓	✓	✓						

**Table 2 sensors-25-01992-t002:** Object action affordance description. Symbol “✓” represents that the action has some relations with the part of an object.

	*grasp*	*pourWith*	*placeOn*	*push*	*press*	*cutWith*	*poundWith*	*brushWith*	*scoopwith*	*insertInTo*
*holdingPart*	✓			✓						
*poundingPart*	✓			✓			✓			
*cuttingPart*	✓			✓		✓				
*scoopingPart*	✓			✓					✓	
*containingPart*		✓								✓
*buttonPart*					✓					
*brushingPart*	✓			✓				✓		
*supportingPart*			✓	✓						

**Table 3 sensors-25-01992-t003:** Semantic representation of object task affordance.

Aspect	Semantic	Mass
$\Theta_{S_{1}}$	*Task(X, use, O_1_, O_2_)*	$m_{s_{1,1}}$
$\Theta_{S_{2}}$	*Task(X, pourOut, O_1_, O_2_)*	$m_{s_{2,1}}$
$\Theta_{S_{3}}$	*Task(X, pourIn, O_1_, O_2_)*	$m_{s_{3,1}}$
$\Theta_{S_{4}}$	*Task(X, grasp, O)*	$m_{s_{4,1}}$
$\Theta_{S_{5}}$	*Task(X, press, O)*	$m_{s_{5,1}}$
$\Theta_{S_{6}}$	*Task(X, insertIn, O_1_, O_2_)*	$m_{s_{6,1}}$
$\Theta_{S_{7}}$	*Task(X, place, O_1_, O_2_)*	$m_{s_{7,1}}$
$\Theta_{S_{8}}$	*Task(X, push, O)*	$m_{s_{8,1}}$
$\Theta_{S_{9}}$	*Task(X, pass, O)*	$m_{s_{9,1}}$

**Table 4 sensors-25-01992-t004:** Semantic representation of object action affordance.

Aspect	Semantic	Mass
$\Theta_{F_{1}}$	*grasp(X, O, part(O))*	$m_{f_{1,1}}$
$\Theta_{F_{2}}$	*push(X, O, part(O))*	$m_{f_{2,1}}$
$\Theta_{F_{3}}$	*press(X, O, part(O))*	$m_{f_{3,1}}$
$\Theta_{F_{4}}$	*cutWith (X, O, part(O))*	$m_{f_{4,1}}$
$\Theta_{F_{5}}$	*scoopWith (X, O, part(O))*	$m_{f_{5,1}}$
$\Theta_{F_{6}}$	*pourWIth(X, O, part(O))*	$m_{f_{6,1}}$
$\Theta_{F_{7}}$	*insertInTo(X, O, part(O))*	$m_{f_{7,1}}$
$\Theta_{F_{8}}$	*brushWith (X, O, part(O))*	$m_{f_{8,1}}$
$\Theta_{F_{9}}$	*poundWith (X, O, part(O))*	$m_{f_{9,1}}$
$\Theta_{F_{10}}$	*placeOn (X, O, part(O))*	$m_{f_{10,1}}$

**Table 5 sensors-25-01992-t005:** Semantic representation of action sequence.

Aspect	Semantic	Mass
$\Theta_{A_{1}}$	*X grasp part(O)*	$m_{a_{1,1}}$
$\Theta_{A_{2}}$	*X push part(O)*	$m_{a_{2,1}}$
$\Theta_{A_{3}}$	*X press part(O)*	$m_{a_{3,1}}$
$\Theta_{A_{4}}$	*X grasp part(O) pourWith part(O)*	$m_{a_{4,1}}$
$\Theta_{A_{5}}$	*X grasp part(O) cutWith part(O)*	$m_{a_{5,1}}$
$\Theta_{A_{6}}$	*X grasp part(O) poundWith part(O)*	$m_{a_{6,1}}$
$\Theta_{A_{7}}$	*X grasp part(O) brushWith part(O)*	$m_{a_{7,1}}$
$\Theta_{A_{8}}$	*X grasp part(O) scoopWith part(O)*	$m_{a_{8,1}}$
$\Theta_{A_{9}}$	*X grasp part(O_1_) placeOn part(O_2_)*	$m_{a_{9,1}}$
$\Theta_{A_{10}}$	*X grasp part(O_1_) insertInTo part(O_2_)*	$m_{a_{10,1}}$
$\Theta_{A_{11}}$	*X grasp part(O_1_) pourWith part(O_1_)*	$m_{a_{11,1}}$

**Table 6 sensors-25-01992-t006:** Objects and related tasks involved in the experiment. Symbol “✓” represents that the task has some relations with the object.

	Chair	Remote	Mug	Knife
*pass*		✓	✓	✓
*use*				✓
*pourOut*			✓	
*grasp*		✓	✓	✓
*place*	✓	✓	✓	✓
*push*	✓	✓	✓	✓
*press*		✓		
*pourIn*			✓	
*insertIn*			✓	

**Table 7 sensors-25-01992-t007:** Records of tasks performed on object in the user’s daily life for single object.

	*pass* *-mug*	*pourOut-* *mug*	*grasp-* *mug*	*place-* *mug*	*push-* *mug*	*pourIn-* *mug*	*insertIn-* *mug*
1	1	0	0	0	0	0	0
2	0	1	0	0	0	0	0
3	0	0	0	0	0	1	0
…	…	…	…	…	…	…	…
108	1	0	0	0	0	0	0
109	1	0	0	0	0	0	0
110	0	1	0	0	0	0	0

**Table 8 sensors-25-01992-t008:** Records of tasks performed on object in the user’s daily life for multiple objects.

	1	2	…	109	110
*Task(X, use, knife, table)*	0	0	…	1	0
*Task(X, place, knife, table)*	1	1	…	0	1
*Task(X, insertIn, bottle, mug)*	0	1	…	0	0
*Task(X, pourOut, bottle, mug)*	1	0	…	1	0
*Task(X, pourIn, bottle, mug)*	0	0	…	0	1

**Table 9 sensors-25-01992-t009:** Consistent configuration between task and action intention for single object. Symbol “✓” represents that the task/part has some relations with the object/action respectively.

	Object	Chair		Remote		Mug		Knife			
Task											
*pass*					✓	✓		✓		*grasp*	
*use*								✓		*cutWith*	
*pourOut*						✓				*pourWith*	
*grasp*					✓		✓		✓	*grasp*	
*place*		✓			✓		✓		✓	*placeOn*	
*push*			✓		✓		✓		✓	*push*	
*press*				✓						*press*	
*pourIn*						✓				*pourWith*	
*insertIn*						✓				*insertInTo*	
		*supporting* *Part*	*holding* *Part*	*button* *Part*	*holding* *Part*	*containing* *Part*	*holding* *Part*	*cutting* *Part*	*holding* *Part*		Action
										Part	

**Table 10 sensors-25-01992-t010:** Consistent configuration between task and action sequence intention for single object.

	Chair	Remote	Mug	Knife
*pass*		*r^11^*	*r^10^*	*r^7^*
*use*				*r^2^*
*pourOut*			*r^5^*	
*grasp*		*r^12^*	*r^12^*	*r^12^*
*place*	*r^29^*	*r^30^*	*r^30^*	*r^30^*
*push*	*r^20^*	*r^20^*	*r^21^*	*r^23^*
*press*		*r^26^*		
*pourIn*			*r^19^*	
*insertIn*			*r^27^*	

**Table 11 sensors-25-01992-t011:** Consistent configuration between task and action sequence intention for multiple objects.

Task	Rule
*Task(X, use, knife, table)*	*r^30^*
*Task(X, place, knife, table)*	*r^28^*
*Task(X, insertIn, bottle, mug)*	*r^27^*
*Task(X, pourOut, bottle, mug)*	*r^18^*
*Task(X, pourIn, bottle, mug)*	*r^19^*

**Table 12 sensors-25-01992-t012:** Ablation experiment configurations and results. “✓” denotes that the model contains this module.

Experiment	OA	OT	D-S	Avg. Number of Interactions (Single Object)	Avg. Number of Interactions (Multi-Objects)
E1	✓			1.775	1.350
E2		✓		3.225	4.733
E3	✓	✓	✓	1.525	1.350

## Data Availability

Restrictions apply to the datasets.

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
