# Peer review of "Intention Reasoning for User Action Sequences via Fusion of Object Task and Object Action Affordances Based on Dempster–Shafer Theory"

_sensors, 2025, doi:10.3390/s25071992_

Round 1

Reviewer 1 Report

Comments and Suggestions for Authors

In this paper is proposed an intention reasoning method for users’ action sequences by fusing object-task and object-action affordances based on D-S Theory.

The proposal is composed of different stages.

Firstly, authors use an object recognition network to obtain object’s class in which the user is focused, then establish the object/task ontology and object-task affordance. Authors encode the object-task affordance into probabilistic relations based on CP-logic to build the object-task intention reasoning model.

Reinforcement learning is used to update user’s habits.

To accurately infer user intention, authors fusion object-task and object-action affordance.

Then, the D-S evidence theory is used to reason about both task intentions and action intentions under the dual constraints of object-task and object-action affordance.

As presented by the authors the proposed methodology allows the WMRA to accurately understand users’ task intentions and select and execute appropriate actions.

On section 3 authors describe methods used, particularly on section 3.3 authors describe in detail the DS Model used, except for some minor details, the section it’s very well described, and let readers to understand the proposed approach.

 It is desirable to equalize paper sections, as for example section 2 it is very short in relation to others.

In general, the paper is correctly wrote and self-contained, however, are required more details about experiments (section 4), how many experiments were made on the training model? (Lines 498-499). Authors mention at lines 512 and 513 “excellent segmentation performance”, but it is required to mention how this performance was measured, a table or graphs could help a lot.

Authors report experiments on two users, however it is not clear if this number of users is enough to conclude that the proposed methodology can adapt to different habits of different persons, experiments with more people will be suitable.

Authors claim that their method reduce the number of interactions for single target objects, how this was measured?

In order to accept this affirmation authors should present some comparisons.

Additionally, some other minor issues should be solved before acceptation.

In line 30 please provide the correct words for the WMRA acronym, as presented on abstract. Acronym STGCN-LSTM should be defined and should be written as ST-GCN-LSTM.

A blank space is required between “.” and “In” on line 52.

Paragraph on lines 120-124 should be rewritten, as in the form it’s presented the first sentence it’s confusing.

There is an isolated  “t”  on line 133.

At line 134, should be referenced also the section 5 Conclusions.

Sentence on lines 136 to 137 should be rewritten to make a more fluid reading.

CRF on line 141 hasn’t been defined.

Please verify captions at right column of Figure 1.

A period is missing on line 153.

At line 172 and 173 it better to use “object’s class” than “object’s name”, similarly, at line 173, “laser pointer”, could be used instead “laser interaction” as those words are more appropriated in the context.

At line 192 it is required a reference to the “International Classification of Functioning, Disability, and Health guidelines”.

At line 282 its preferred to use “equation” to “formula”, and please verify the position of “n” superscript on both sigma.

WMRA on 348 do not require to be defined as it has been already defined at the beginning of introduction.

At line 358 m() function should be defined.

Line 370 should be written without indentation.

Theta letters at lines 374 and 375 are different, please standardize their use, as well for table 3.

Letter “n” at line 501 should be in italics (math).

Text inside graphs of Figures 7, 8 and 10 must be increased for a better reading.

References

Standardize the use of “.” And not “,” at the end of article titles, and also avoid the use of “” for titles as in reference [38].

In reference [25], there is a “IEEE” before the paper title.

The following references has been repeated, [27] is the same as in [8] and in [3], and [26] is the same as [7].

Reviewer 2 Report

Comments and Suggestions for Authors

In the context of wheelchair with a robotic arm, the authors investigated an intention reasoning method for users’ action sequences by fusing object-task and object-action affordances based on D-S Theory.  In this paper they introduce their intent reasoning framework, their object-task and object-action affordance methods, and they validate it experimentally. They used multi-label classification and detailed a multi-object inference of action sequence intentions.

Please find below some comments.

First about the context of a wheelchair with a robotic arm, It is used as a general background and to define object and tasks sets, but the robotic model/control law/motion did not appear in this paper. So, it is a little bit disaponting that nothing is said about how the robot will behave after having identified the action.

In Fig.2, you detailed a lot the different Object ontology where you gathered elements into class. However, I think more information about the importance of class might be useful. For example, why is it important to have a pour class that gathers pourIn and pourOut? Why did you stay at the container or tool class but not at a more detailed class?

In l.542, you said that you used user feedback, but you did not explain how you did it.

In line 558, you said that the results indicate that the model suggest that both subjects have similar habits regarding the use of chair. May you can give more information about the subject (same family or same institution?, How long have they been in a chair, same environment, etc.). I totally agree that all information could not be given to protect them, but some could change the conclusion of the analysis.

In Figure 7., Adding #Subject 1 or 2 in the graph title could help the reading. May a more factorized title could also help to read it.

The texts of the figures are too small. Ex. Fig3. Figure big enough to use a bigger front.

In Figure 8. Formalism of Subject#1 and #2 is not the same than in the entire article.

From l.604 to 607, you concluded that your action intention reasoning method can effectively adapt to different user habits, but you also say that the probability curves are relatively stable, indicating that the two participants have similar lifestyles. From those two points, I do not think that we can conclude on the adaptability of your action intention reasoning method on someone that would not have the samelife style. It could have been interesting to validate it with a 3rd subject whose lifestyle is completely different to see if you have the same results.

In Fig 10, legend and axis label are too small. Caption is hard to read and understand the difference between the different curves.

Discussion about the learning rate that can balance between speed and enhance the model’s ability.

You showed results in figure 11, but I did not understand what the learning rate is. Maybe more details on the setup of the algorithm could be useful.

In 4.4, you are performing several series of experiments, but you did not explain this point. How many experiments? Is it statistically representative? Which subject did you use? You showed average value with max/min (I suppose), but how did you compute the mean value?

In table 12, If I well understood what you want to show, I think two cases are lacking OA-DS and OA-OT such as we can clearly this the improvement from your proposed algorithm. Considering it, the conclusion from Fig13 might change even if what you show is still good.

In Fig13. You showed that for multi-object constraints, the average number of iteractions, but also the min/max are the same. If you can add more discussion, I think this point is quite interesting. To compare also the usability of the OT, OA and your method, maybe other characteristics such as the training time could be considered.

Typo:
- Upper case for words from line l151 to l.160.
- I think that there is a mistake in the sentence l.295 “can not only perform atomic actions .. but also can perform atomic actions...”.
- upper case l.343.
- l555. In the previous.

Reviewer 3 Report

Comments and Suggestions for Authors

1. Content of the Article:
The paper introduces an innovative intention reasoning method for user action sequences based on object-task and object-action affordances using Dempster-Shafer Theory. While the overall structure is logical, the lack of detailed experimental results and a clear conclusion section significantly weakens the paper’s persuasiveness. It is essential to include concrete data from real-world applications or simulations to validate the effectiveness of the proposed method. Additionally, the semantic representation of action sequences (e.g., Table 5) could benefit from more detailed explanations to ensure clarity for readers unfamiliar with this concept.
2. Research Approach and Methodology:
The research methodology is reasonably structured, but there is a need for further elaboration on how the affordance models are integrated into the intention reasoning process. Specifically:
A step-by-step breakdown of how object-task and object-action affordances are fused using Dempster-Shafer Theory would enhance readability.
More details on the selection of eleven binary affordance aspects (e.g., why these specific aspects? How were they validated?) would strengthen the methodological rigor.
The discussion on “semantic constraint rule models” is vague; formalizing these rules and providing examples would improve understanding.
3. Innovation:
While the paper proposes a novel fusion framework for intention reasoning, it lacks sufficient comparison with existing methods in the literature. To better highlight its innovation:
A comprehensive review of related work in intention inference and affordance-based robotics should be provided.
The advantages of the proposed method over existing approaches (e.g., computational efficiency, accuracy) should be explicitly stated and validated through experiments.
4. Language and Description:
The language is generally clear, but there are instances where explanations could be more precise:
For example, the sentence “the method not only integrally considers the user’s operating habits, but also fully utilizes the action attributes of the object’s functional regions” could be rephrased for better clarity and conciseness.
Technical terms like “affordance” and “Dempster-Shafer Theory” should be briefly defined or referenced when first introduced to ensure accessibility for a broader audience.
5. Literature References:
The paper’s reference list appears incomplete, as key citations on Dempster-Shafer Theory and CP-Logic are missing (e.g., foundational works by Zadeh, Shafer, and Yager). To improve this section:
A complete bibliography should be provided, including seminal papers in the field.
Additional references to recent studies on affordance-based robotics or intention inference would demonstrate a deeper engagement with the existing literature.
6. Overall Evaluation:
The paper demonstrates potential for advancing intention reasoning in robotic systems, but significant improvements are needed:
Major Revisions (Large Revision): Focus on:Providing detailed experimental validation to demonstrate the method’s effectiveness.
Clarifying and expanding explanations of key concepts (e.g., semantic constraint rules, affordance aspects).
Enhancing the literature review and references to strengthen the theoretical foundation.
Polishing the language for improved readability and conciseness.
7.Final Recommendation: The paper requires substantial revisions to address these issues before it can be considered for publication.

Comments on the Quality of English Language

Average

Round 2

Reviewer 2 Report

Comments and Suggestions for Authors

First of all, I want to thank the authors for their meticulous answers.

Considering the previous version, authors added more information about some simulations they performed. They added also a 3rd subject to demonstrate more the capability of their proposed algo.

Further comparison with others algorithm were detailed that bring more depth to the results.

Typo:
- graph (l) Fig.7 title should be #3.
- table 7 Two columns with “push-mug”.

Reviewer 3 Report

Comments and Suggestions for Authors

No problem

Comments on the Quality of English Language

Average
